# Modeling and Experimental Analysis of Tofu-Drying Kinetics

Cécile Hanon, Morouge Al Hassan, Soulaimane Nassouh, Salahaldin Abuabdou [ID], Charlotte Van Engeland and Frédéric Debaste * [ID]

Transfers, Interfaces and Processes (TIPs) Department, Brussels School of Engineering, Université libre de Bruxelles (ULB), Avenue F.D. Roosevelt, 50 CP165/67, 1050 Bruxelles, Belgium; cecile.hanon@ulb.be (C.H.); morouge.al.hassan@ulb.be (M.A.H.); soulaimane.nassouh@ulb.be (S.N.); salahaldin.abuabdou@ulb.be (S.A.); charlotte.van.engeland@ulb.be (C.V.E.)
* Correspondence: frederic.debaste@ulb.be

**Featured Application**

This research supports the optimization of drying processes and enhances predictive modeling in food cooking applications.

**Abstract**

Drying critically shapes tofu's texture, structure, and final appearance, whether it occurs during cooking or is applied intentionally in reprocessing. This study aimed to characterize the drying kinetics of tofu through experimental analysis and through modeling. Tofu samples were dried at temperatures ranging from 40 °C to 90 °C using a convective drying tunnel and an oven. Shrinkage and color changes were analyzed. Empirical models, a shrinking-core model and a newly developed oven-cooking model were tested against experimental data. The drying kinetics exhibited a constant and a decreasing rate phase, which were separated by a water content threshold of 2.56 $kg_W/kg_{DS}$. Tofu undergoes non-enzymatic browning and exhibited total shrinkage of 0.38. These physical changes were more significant at lower drying temperatures when the product was dried below a water content of 1.39 $kg_W/kg_{DS}$. The logarithmic model provided the best fit ($R^2 \geq 0.9920$) to the experimental data. However, the cooking model shows good results as well ($R^2 = 0.9678$) and offers physical validity. This study provides evidence that the drying mechanisms of tofu are not temperature-dependent within the studied range. It also emphasizes the importance of drying time over drying temperature in the physical changes of the product. The successful fit of the cooking model highlights the link between drying and cooking processes, suggesting potential applications in both areas.

**Keywords:** tofu; drying; heat and mass transfer; modeling; cooking

## 1. Introduction

Tofu is a traditional Chinese food product made from curdled soy milk. Its consumption has become widespread due to its nutritional value, health benefits, and suitability as an alternative to animal protein [1–4]. Tofu exists in a wide variety of forms, differentiated by water content. Silken tofu, also known as soft tofu, has a water content of about 87–90%, while firm tofu is pressed to reduce its moisture content, resulting in a more compact matrix with a water content of about 67–81% [5]. Finally, leisure-dried tofu is a reprocessed form of tofu that is dried before marinating to enhance its flavor [6]. Its production usually involves an extended drying phase of three hours at an elevated temperature of around 85 °C [7].

Drying is, therefore, a possible transformation process in the production and consumption of tofu. It occurs both in domestic cooking processes, like oven cooking, where it is one of the key processes [8], and in industrial settings like the production of leisure-dried tofu [7]. In particular, leisure-dried tofu has become increasingly popular on the market due to its taste, ready-to-eat attributes, and nutritious advantages [6,7,9]. Drying tofu changes its flavor and texture, creating a product that is crispy yet tender, and extends storage time by reducing the risk of spoilage [10]. Therefore, studying the drying kinetics of tofu has direct industrial applications.

While water content is a key factor in tofu texture, it also influences other physical properties, such as color and shape [11]. Significant water loss inevitably leads to shrinkage and impacts the product's coloration [12,13]. Since color and shape influence consumer acceptance [12], further examination of their relationship to the drying process is necessary.

Baik and Mittal [14] investigated the thermal properties of tofu as a function of water content, but the drying process itself has not been studied. The drying kinetics of soybean grains [15] and okara (a byproduct of soy milk and tofu production) [16–18] have been studied and characterized. Several empirical models, such as Page, Henderson, and Thompson, have been tested for soybean drying and have produced satisfactory results [15]. For okara, both empirical models [16] and physical models (heat and mass transfer [18]) have been successfully tested. While the drying kinetics of other soy-related products have been studied, this has not been the case for tofu itself.

In this study, the experimental results were compared to three types of models: simple empirical models [19], a shrinking-core model (SK) [20], and a newly developed phenomenological model for oven-cooking processes (the cooking model). Mathematical modeling of tofu drying kinetics serves multiple purposes. First, it helps us understand and predict the physical mechanisms involved in the drying process. Second, the two physics-based models examined here provide meaningful parameters that can inform process design [20,21]. Finally, this study evaluates a newly developed model for oven cooking. By linking drying dynamics to cooking processes through mathematical modeling, this study sheds light on the key mechanisms that influence cooking performance [22].

Therefore, the objective of this work is to characterize the drying kinetics of tofu at different temperatures using experimental analysis and modeling. Changes in shape and color during the drying process were also examined.

## 2. Materials and Methods

*2.1. Experimental Methods*

2.1.1. Tofu Samples

The tofu used in this study was plain, firm tofu, packaged in 250 g portions. It was purchased from a local supermarket (brand: Carrefour Bio, Brussels, Belgium). The outermost compressed layer formed during processing was removed from each fresh tofu block before it was cut into 8 cm$^3$ cubes (2 cm edges). The nutrition facts for 100 g of the selected tofu are as follows: 8.1 g of fat, 2.0 g of carbohydrates, <0.5 g of dietary fiber, 12.0 g of protein, and <0.01 g of salt.

2.1.2. Drying Process

Drying experiments were conducted using two types of convective drying systems: a drying oven and a drying tunnel.

The thermal degradation of bioactive compounds in food occurs under different drying conditions, depending on the compound being studied [23,24]. Therefore, the experiments covered a wide range of temperatures, starting at 40 °C (higher than the usual ambient temperature), and ending at 90 °C (below the boiling point of water). Drying-oven

experiments were performed using a Memmert UBF 400 unit (Memmert GmbH + Co. KG, Büchenbach, Germany), at 40, 50, 60, 70, 80, and 90 °C. The samples were weighed using a Sartorius Entris BCE2201I-1S balance (Sartorius AG, Göttingen, Germany) at the following time points: 0, 10, 20, 30, 40, 50, 70, 90, 120, 150, 180, 210, 240, and 270 min. All tests were conducted in triplicate.

Additional drying experiments were performed in a drying tunnel at temperatures of 40, 50, and 60 °C. The maximal achievable temperature for this setup was 60 °C. This apparatus allows for continuous weight monitoring (using a Sartorius Entris BCE2201I-1S) without interrupting the drying process and provides well-controlled drying conditions. Additionally, a Nikon D3100 camera (Nikon Corporation, Tokyo, Japan) (4608 × 3072 px) was positioned above the sample to acquire images and monitor physical changes during drying. This setup was described by Spreutels et al. [25] and is illustrated in Figure 1. The experiments were conducted with an airflow of 20 Nm³/h, and the mass and images were recorded every five minutes.

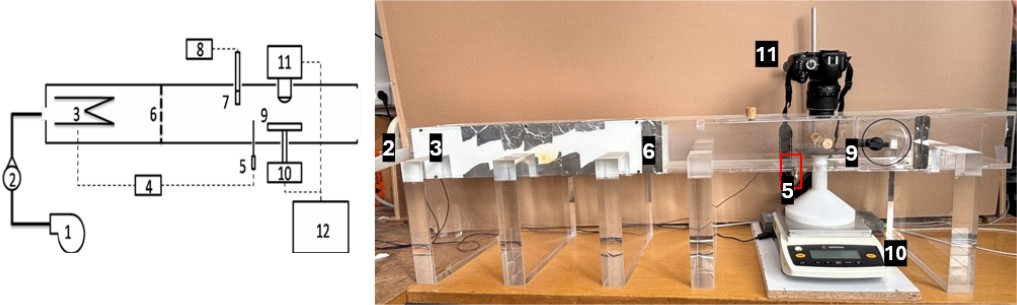

**Figure 1.** Tunnel dryer: (1) fan, (2) rotameter, (3) heating device, (4) heating controller, (5) control thermocouple, (6) gas diffuser, (7) hygrometer probe (not used here), (8) hygrometer acquisition (not used here), (9) sample holder, (10) scale, (11) imaging device, and (12) acquisition computer, from [25].

After each drying test, the dry mass of the sample $m_{DS}$ was determined by placing it in a drying oven at 102 °C for three days until it reached a stable weight.

The average moisture content in dry basis, $X_i$ (*i*th measure), was determined from the measured mass values $m_i$ of the samples:

$$X_i = \frac{m_i - m_{DS}}{m_{DS}} \tag{1}$$

The drying rate $j$ was determined as follows:

$$j = -\frac{dX}{dt} \tag{2}$$

For a better comparison between the experiments, the values of $X$ and $j$ were normalized, respectively, by the initial moisture content, $X_0$, and the maximum evaporation rate, $j_{max}$:

$$X^* = \frac{X}{X_0} \tag{3}$$

$$j^* = \frac{j}{j_{max}} \tag{4}$$

2.1.3. Color and Size Variations

Images acquired in the drying tunnel were analyzed for color and size variation. The mean color was extracted from a fixed region (approximately 650 × 650 px), and color changes $\Delta E_i$ were calculated in the CIELAB color space ($L^*$ for lightness, $a^*$ for redness, and $b^*$ for yellowness [26]):

$$\Delta E_i = \sqrt{(L_0^* - L_i^*)^2 + (a_0^* - a_i^*)^2 + (b_0^* - b_i^*)^2} \tag{5}$$

To track the size of the sample, a grayscale threshold was applied to separate the sample from the background. Then, the projected area was estimated based on the pixel count and the image scale [27]. Extrapolating these results under the assumption of isotropic shrinkage yields a deduced volume and a total shrinkage $S$, which is the ratio of the final to the initial volume [13]:

$$S = \frac{V_{final}}{V_0} \tag{6}$$

The size variations were compared to a shrinkage model developed by Bukamba Tshanga et al. [28]. This model describes how the relative volume of the product changes with water content and a fraction, $\alpha$, representing the lost liquid volume that is replaced by gas:

$$\frac{V_i}{V_0} = 1 - \epsilon_0(1 - \alpha)(1 - X_i^*) \tag{7}$$

where $\epsilon_0$ represents the initial porosity of the product, which was estimated as the ratio of the initial water volume present in the product (obtained from the experimental dry mass) to the initial volume of the product. The only parameter adjusted to the experimental data during the fitting process was $\alpha$. Since the model considers $\alpha$ to be constant, it is linear.

### 2.1.4. Data Processing

The drying rate $j$ and color data $\Delta E$ were smoothed using a Savitsky–Golay filter with a third-order polynomial and a window size of 13 points.

### 2.2. Mathematical Models

Six mathematical models of varying complexity were evaluated against experimental data: (i) four empirical models frequently used in drying kinetics [19,29], (ii) a physical model based on one-dimensional mass transport equations (the shrinking-core model (SK)) [20], and (iii) a newly developed heat and mass transfer model that simulates oven-based cooking processes (the cooking model, or the HMT model for abbreviation).

To evaluate the suitability of the models, two criteria were used: $R^2$ and root-mean-square error (RMSE):

$$R^2 = 1 - \frac{\sum_{i=1}^{N}(y_{i,exp} - y_{i,pred})^2}{\sum_{i=1}^{N}(y_{i,exp} - \bar{y}_{exp})^2} \tag{8}$$

$$RMSE = \frac{\sqrt{\sum_{i=1}^{N}(y_{i,exp} - y_{i,pred})^2}}{N} \tag{9}$$

For all models, the fitted parameters were computed using the least squares method.

### 2.2.1. The Empirical Models

The selected empirical models are widely used in the literature to model food-drying kinetics [19,29]. The four chosen models are presented in Table 1. The parameters for these models were fitted to the relative moisture content, *X*\*, obtained experimentally (one fitting per experimental condition).

### 2.2.2. The Shrinking-Core Model

This model was initially developed to describe the process of air-drying yeast in a fluidized bed [20]. The parameters and geometry were adapted to the oven-drying process. The balance equations remain unchanged:

$$\frac{dX}{dt} = -j \tag{10}$$

$$V_{oven}\rho_A \frac{dY}{dt} = Q(Y_{in} - Y) + m_{DS}j \tag{11}$$

$$Q(Y_{in} - Y)L_{vap} + QC_{p_A}(T_{in} - T) = m_{DS}(C_{p_{DS}} + C_{p_W}X)\frac{dT}{dt} + m_{DS}C_{p_W}(T - T_{in})\frac{dX}{dt} \tag{12}$$

Equation (10) is the balance equation for the water content in the solid ($X$) and is governed by the drying rate $j$. Equation (11) is the balance equation for the water content in the air of the convective drying oven ($Y$). It is governed by the volumetric flow rate of air entering the oven ($Q$) and the water evaporating from the product. Finally, Equation (12) is the balance equation for the global enthalpy of the system. On the left-hand side, the first term represents the energy carried by vapor due to phase change (with $L_{vap}$ being the latent heat of vaporization). The second term represents the sensible heat carried by the oven air. The right-hand side of the equation is composed of two terms: the first term represents the heat required to change the temperature of the solid and its moisture, and the second term represents the energy associated with moisture variation.

**Table 1.** Selected empirical mathematical models.

| Model Name | Model | Fitted Parameters |
|---|---|---|
| Newton | $X^* = \exp(-kt)$ | $k$ |
| Page | $X^* = \exp(-kt^n)$ | $k, n$ |
| Henderson–Pabis | $X^* = a\exp(-kt^n)$ | $a, k, n$ |
| Logarithmic | $X^* = a\exp(kt) + c$ | $a, k, c$ |

These balance equations are coupled with a model that expresses $j$, allowing the drying process to be described as two distinct consecutive periods. These periods are separated by a critical water content $X_{crit}$, which is determined experimentally. The constant drying-rate period (phase I, when $X > X_{crit}$) is followed by a falling-rate period (phase II, when $X < X_{crit}$). For phase I, the drying rate $j_I$ is expressed as follows:

$$j_I = k_{SK}a\left[C_{sat}(T) - \frac{P}{R_g T}\frac{Y}{\frac{1}{M_A} + \frac{Y}{M_W}}\right] \tag{13}$$

$a$ is the solid's specific surface area, and $k_{SK}$ is the external mass transfer coefficient. The driving force is the difference between the saturation concentration ($C_{sat}$) and the actual vapor concentration. $C_{sat}(T)$ follows the Clapeyron law [20].

For phase II, the drying rate $j_{II}$ is expressed as follows:

$$j_{II} = \frac{a}{\frac{1}{k_{SK}} + \frac{1}{k_{intern}}}\left[C_{sat}(T) - \frac{P}{R_g T}\frac{Y}{\frac{1}{M_A} + \frac{Y}{M_W}}\right] \tag{14}$$

During phase II, an internal resistance to mass transfer becomes a governing factor [20]. $k_{intern}$ represents the internal mass transfer coefficient:

$$k_{intern} = \frac{D_{eff}}{L - L_{int}} = \frac{D_{eff}}{L(1 - \frac{X - X_{res}}{X_{crit} - X_{res}})} \tag{15}$$

with $L$ as the specific length of the sample and $X_{res}$ as the residual water content. $D_{eff}$ is the effective diffusion coefficient:

$$D_{eff} = \frac{\epsilon_{SK}}{\tau_{SK}}D = \xi D \tag{16}$$

In this model, porosity ($\epsilon_{SK}$) and tortuosity ($\tau_{SK}$) are considered constant. The ratio between porosity and tortuosity is represented by the variable $\xi$.

The fitting method for this model involved adjusting $k_{SK}$ and $\xi$ to the entire dataset, thereby including all experimental conditions within a single fitting procedure. Fitting was performed on the experimental drying rate $j$ obtained from $X$, as stated in Equation (2). The main parameter values are presented in Section 3.3. The complete set of parameter values is available in Appendix A.

### 2.2.3. The Cooking Model

Cooking is the deliberate alteration of food products through the transfer of heat, which changes the products' water content, mainly through evaporation. Therefore, drying is a key mechanism in controlling cooking, making it interesting to compare experimental data obtained for the drying process with a heat and mass transfer model developed for oven cooking. This one-dimensional model evaluates the local temperature and water content of the product during the process of oven cooking. The modeled system is represented in Figure 2.

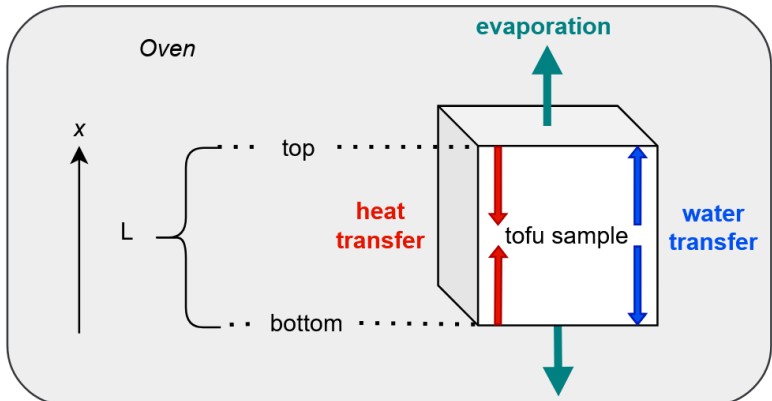

**Figure 2.** Diagram illustrating the system configuration as represented by the cooking model.

Mass transfer is driven by diffusion:

$$\frac{\partial X}{\partial t} = D_{eff}(X)\frac{\partial^2 X}{\partial x^2} \tag{17}$$

The effective diffusion coefficient $D_{eff}$ considers both the molecular ($D$) and the capillary ($D_{cap}$) diffusion of water in the food matrix. $D_{cap}$ is expressed as an empirical correlation depending on water content [30]. The developed expression of $D_{eff}$ is as follows:

$$D_{eff} = D + D_{cap} = D + (1+B)Z\exp\left(A + BX\right) \tag{18}$$

The heat transfer balance equation is as follows:

$$\frac{\partial T}{\partial t} = \lambda \frac{V}{m_{DS}(C_{p_{DS}} + C_{p_W}X)}\frac{\partial^2 T}{\partial x^2} + D_{eff}(X)\frac{C_{p_W}}{C_{p_{DS}} + C_{p_W}X}\frac{\partial X}{\partial x}\frac{\partial T}{\partial x} \tag{19}$$

The first term on the right-hand side corresponds to the conductive term. The second term of the equation represents the energy convected by water transport in the product and depends on the gradients of both the water content and the temperature.

$\lambda$ is the thermal conductivity of the product in W/m/K and is a weighted sum of the thermal conductivities of water and of the dry product:

$$\lambda = \left(\frac{X}{1+X}\right)\lambda_W + \left(1 - \frac{X}{1+X}\right)\lambda_{DS}, \tag{20}$$

The boundary condition for mass transfer is

$$\left.\frac{\partial X}{\partial t}\right|_{x=0 \text{ or } x=L} = -\frac{\Omega}{V}D_{eff}(X)\frac{\partial X}{\partial x} - j_{HMT} \tag{21}$$

including diffusion (the first term on the right-hand side) and evaporation (the second term on the right-hand side). $j_{HMT}$ is the evaporation rate, which is specific to the cooking model:

$$j_{HMT} = k_{HMT}\frac{\Omega}{m_{DS}}(C_{int} - C_A) \tag{22}$$

$k_{HMT}$ is the mass transfer coefficient in m/s. Evaporation is driven by the difference in vapor concentration at the interface ($C_{int}$) and in the air ($C_A$). $C_{int}$ is expressed as follows:

$$C_{int} = \frac{A_W(X)P_{sat}(T)M_W}{R_g T} \tag{23}$$

To estimate $C_{int}$, the water activity $A_W$ is calculated from the water content using the BET model, which describes desorption isotherms and relates $X$ to $A_W$ [31]:

$$X = \frac{A_W K_{BET}}{(1 - A_W)(1 + (K_{BET} - 1)A_W)}X_{BET} \tag{24}$$

$K_{BET}$ is the BET constant, and $X_{BET}$ is the monolayer moisture content.

It is assumed that the humidity at the interface corresponds to saturation at the temperature of the system, which is calculated using Clausius–Clapeyron's law [32].

For heat transfer, the boundary condition is given by the following equation:

$$\lambda\left.\frac{\partial T}{\partial x}\right|_{x=0 \text{ or } x=L} = C_{pW}\frac{m_{DS}}{V}D_{eff}(X)\frac{\partial X}{\partial x}(T - T_W) - h(T - T_{in}) - L_{vap}\frac{m_{DS}}{\Omega}j_{HMT} \tag{25}$$

The first term on the right-hand side represents the convective heat carried by moisture diffusion. The second term represents the convective heat loss to the ambient air. The last term represents the latent heat loss due to evaporation.

The model was fitted by adjusting six parameters to the entire dataset, encompassing all experimental conditions in a single fitting procedure. The fitted parameters were the empirical parameters $A$, $B$, and $Z$ from Equation (18); the two parameters from the BET model ($K_{BET}$ and $X_{BET}$); and the thermal conductivity of the dry product, $\lambda_{DS}$. Similar to the shrinking-core model, fitting was performed on the experimental drying rate $j$ (Equation (2)). The main parameter values are presented in Section 3.3. The complete set of parameter values is provided in Appendix A.

## 3. Results

### 3.1. Drying Kinetics

#### 3.1.1. Oven Dryer

The mean initial tofu water content $\bar{X}_0$ was $3.44 \pm 0.26$ kg$_W$/kg$_{DS}$. Figure 3a shows how the relative moisture content $X^*$ changes over time during experiments conducted at different temperatures in the drying oven. Figures for oven-drying experiments are presented as the mean values obtained from triplicates. The individual values can be found in Appendix B and demonstrate the reproducibility of the results.

As expected, the drying rate increases with temperature. Figure 3b shows the relative drying rate $j^*$ as a function of the relative moisture content $X^*$. After normalization, all the curves exhibit the same trend, suggesting that the drying mechanisms are not temperature-dependent within the studied temperature range.

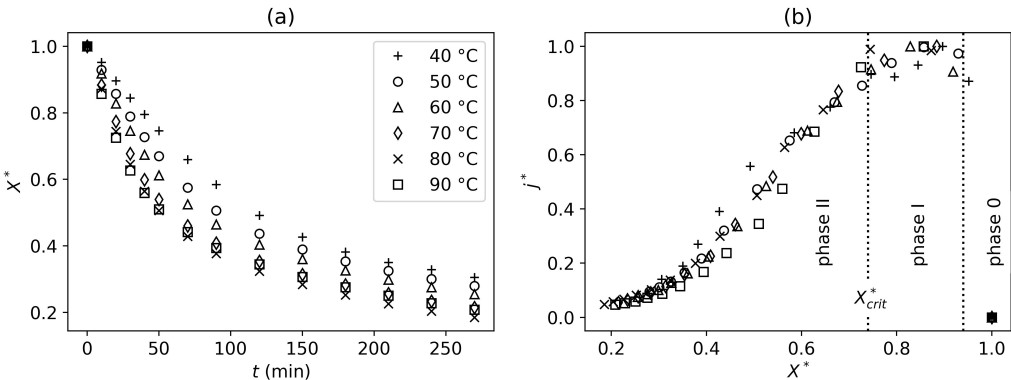

**Figure 3.** (**a**) Evolution of the relative moisture content $X^*$ of tofu samples as a function of drying time for different drying temperatures in a convective drying oven. (**b**) Evolution of the relative drying rate $j^*$ as a function of $X^*$ for the same experiments. Phase 0: warming-up period; Phase I: constant drying period; Phase II: falling-rate period. Each point represents the mean of the triplicate.

Figure 3b also shows that the drying process can be divided into three sequential periods: an initial warming-up period (phase 0), followed by a constant drying-rate period (phase I), and a falling-rate period (phase II) [13]. The falling-rate period begins at $X = X_{crit}$, which was evaluated at 2.56 $kg_W/kg_{DS}$ (corresponding to $X^* = 0.74$). To determine the value of $X_{crit}$, phase II was defined as beginning when the moisture content deviated by more than 10% from the mean value observed during phase I.

These three identified periods are characteristic stages of convective drying [13]. During the constant-rate period, drying is governed by external transport mechanisms, with evaporation of unbound water occurring at the product surface. Once $X_{crit}$ is reached, internal transport mechanisms become rate-limiting [28], leading to a progressive reduction in the drying rate. These internal mechanisms are predominantly diffusion-controlled, involving both molecular diffusion and capillary transport [33].

### 3.1.2. Tunnel Dryer

Appendix C shows a comparison of the drying curves from the tunnel and oven-drying experiments conducted at the same temperatures. The results confirm the conclusion of Bukamba Tshanga et al. [28] that both drying methods result in very similar drying kinetics. Therefore, the tunnel dryer was used to obtain images for analyzing physical variations during drying.

### 3.2. Physical Variations During Drying

Physical variations in size and color were studied during the tunnel-drying process of tofu.

### 3.2.1. Size Variations During Drying

As can be seen in Figures 4 and 5, the drying process substantially reduces the size of the product. After six hours of drying, the total shrinkage $S$ was equal to 0.38 (Equation (6)).

In Figure 5b, a near-linear relationship is observed between normalized surface area, $\Omega^*$, and relative water content, up to the threshold value of $X = 1.39$ $kg_W/kg_{DS}$ (corresponding to $X^* = 0.44$), indicating that shrinkage is strongly governed by water loss during drying. These observations suggest that water removal directly dictates structural deformation in tofu during drying. This means that shrinkage occurs as a continuous, moisture-driven process, likely due to the collapse of the porous matrix. In the literature, linear shrinkage models are usually applied to products where porosity development during drying is considered negligible [12].

The experiments were fitted to the shrinkage model presented in Equation (7) (see Appendix D for fitting figures). The fitting was satisfactory ($R^2 > 0.9677$) for the linear part of the drying process, above the threshold value for $X$. Across nearly all the experiments, the fitted values of $\alpha$ (the fraction of volume replaced by gas following liquid loss) were below $10^{-12}$ and can, therefore, be considered null. This suggests that during the first stage of the drying process, the porous matrix completely collapses, and no void space is preserved where water once resided.

As mentioned previously, the linear relationship between shape and water content only lasts until $X$ reaches $1.39\ \mathrm{kg_W/kg_{DS}}$. After this threshold, shrinkage becomes temperature-dependent and more significant at lower drying temperatures. This demonstrates that long exposure to drying conditions has a greater impact on product shape than temperature does overall in the drying process.

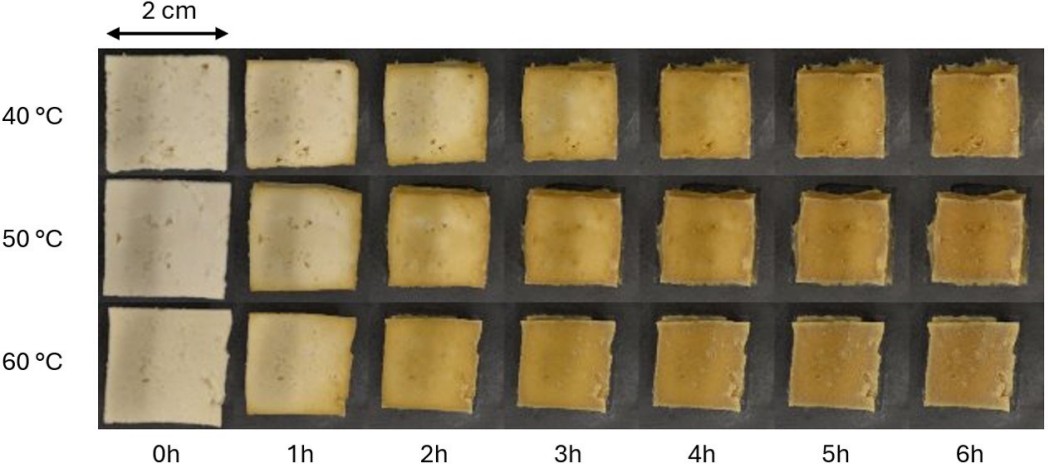

**Figure 4.** Images captured during the convective tunnel drying of tofu samples at different drying temperatures and at different drying times.

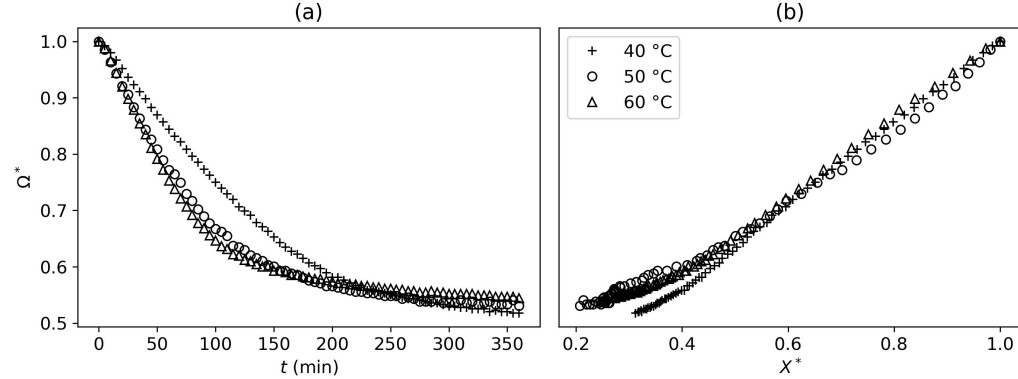

**Figure 5.** (**a**) Evolution of the relative area $\Omega^*$ of tofu samples as a function of drying time in a convective drying tunnel at different drying temperatures. (**b**) Evolution of $\Omega^*$ as a function of relative moisture content of the sample, $X^*$, for the same experiments.

### 3.2.2. Color Variations During Drying

Progressive color changes in tofu samples during drying can be observed in Figure 4. The image analysis enabled us to quantify the evolution of the individual CIELAB components $L^*$, $a^*$, and $b^*$. The graphical results for these components can be found in Appendix E. The observed trends, which are consistent with the visual inspection of the samples, confirm that drying induces a darkening of the surface, accompanied by enhanced red and yellow tones. These changes are characteristic of non-enzymatic browning reactions like Maillard

reactions [34]. The evolution of the three color components $L^*$, $a^*$, and $b^*$ is consistent with trends already identified in the literature for tofu [34].

Figure 6a shows the total color difference $\Delta E$ (Equation (5)) as a function of time. During the early stages of drying, samples exposed to higher temperatures exhibited more pronounced color changes. As drying progressed, $\Delta E$ increased with time, eventually reaching a plateau. Interestingly, Figure 6b shows that, although color change mechanisms appear similar across temperatures at higher moisture contents, more intense discoloration is observed in samples dried at lower temperatures below a certain threshold (of around $X = 1.38 \ \mathrm{kg_W/kg_{DS}}$, corresponding to $X^* = 0.42$). This suggests that longer drying times impact color more than drying temperature.

The thresholds at which color change and size variation become drying-temperature-dependent are quite close (respectively, $X = 1.38 \ \mathrm{kg_W/kg_{DS}}$ and $X = 1.39 \ \mathrm{kg_W/kg_{DS}}$). This means that if the targeted water content is below these thresholds, higher drying temperatures will alter the product less, in terms of both color and shape.

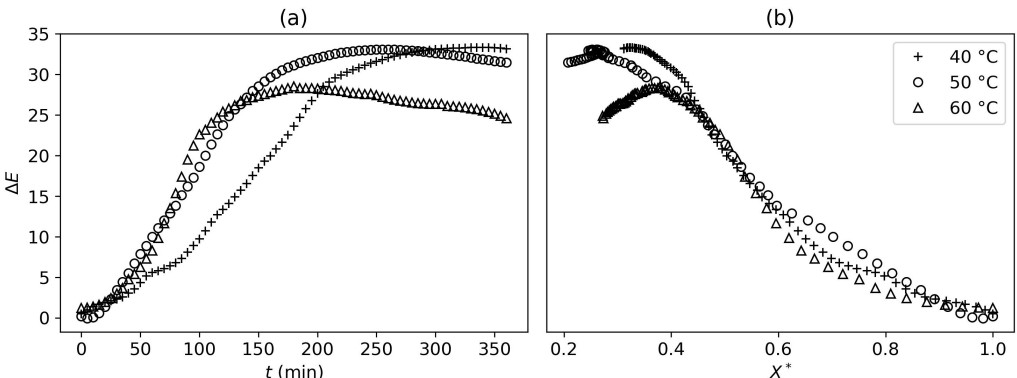

**Figure 6.** (**a**) Evolution of the total color difference $\Delta E$ of tofu samples as a function of drying time in a convective drying tunnel at different drying temperatures. (**b**) Evolution of $\Delta E$ as a function of relative moisture content of the sample, $X^*$, for the same experiments.

### 3.3. Mathematical Modeling

The models presented in Section 2.2 were adjusted to the experimental data from the oven-drying experiments. As discussed in Section 3.1, the kinetics are very similar for the two drying methods. Since the oven allows for a wider temperature range, the models were adjusted to the oven experiments instead of the tunnel-drying experiments.

### 3.3.1. The Empirical Models

The parameter values obtained from the fittings, along with the corresponding plots for the four empirical models, are presented in Appendix F. The logarithmic model produced the best fitting results (Figure 7), followed by the Henderson–Pabis model, the Page model, and finally, the Newton model. Despite the relatively high $R^2$ values obtained, the Newton model does not accurately represent the nonlinear trend of the experimental data. Thus, the Newton model appears to be too simple to describe the more complex drying kinetics of tofu. The other three models provided satisfactory results.

### 3.3.2. The Shrinking-Core Model

For the shrinking-core model, the values and correlations of the parameters are presented in Table 2. The parameters $k_{SK}$ and $\xi$ were optimized to fit the drying rate $j$ obtained from the experimental data. The fitting criteria obtained are as follows: $R^2 = 0.8782$ and RMSE $= 7.795 \times 10^{-6}$. $D_{eff}$ can also be calculated for each drying temperature from the fitting results, ranging from $4.809 \times 10^{-6}$ to $6.537 \times 10^{-6} \ \mathrm{m^2/s}$. Figure 8 shows that overall,

this model is capable of capturing the general drying trends. However, the fit does not perfectly align with the experimental data for all drying temperatures.

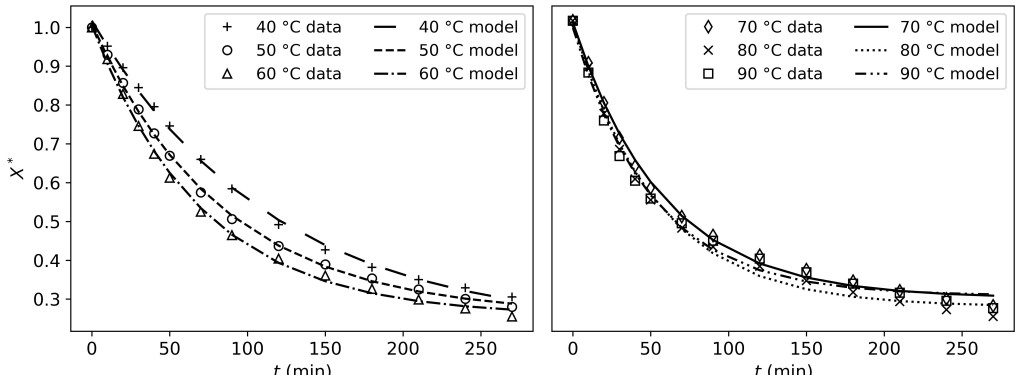

**Figure 7.** Fitting results for the logarithmic model fitted to experimental data obtained at drying temperatures 40 °C–90 °C.

**Table 2.** Parameters used in the shrinking-core model for tofu drying.

| Parameter | Value or Correlation | Unit | Method or Source |
|---|---|---|---|
| $k_{SK}$ | $4.150 \times 10^{-3}$ | m/s | fitted |
| $\xi$ | $1.734 \times 10^{-1}$ | — | fitted |
| $V_{oven}$ | $5.3 \times 10^{-2}$ | $m^3$ | known |
| $X_{crit}$ | 2.56 | $kg_W/kg_{DS}$ | experimental |
| $D$ | $1.87 \times 10^{-10} \times \frac{T_{in}^{2.072}}{P}$ | $m^2/s$ | [35] |
| $C_{p_{DS}}$ | $349.5 \times \left(1 - \frac{X_0}{1+X_0}\right)$ | $J/kg_{DS}/K$ | [12] |
| $Q$ | $2 \times 10^{-4}$ | $kg_A/s$ | estimated |
| $X_{res}$ | 0.00 | $kg_W/kg_{DS}$ | estimated |
| $Y_{in}$ | $1 \times 10^{-3}$ | $kg_W/kg_A$ | estimated |

### 3.3.3. The Cooking Model

For the cooking model, a total of six parameters were fitted to the experimental drying rate *j*. These parameters were optimized to fit all the drying curves simultaneously. The values and correlations of the different parameters are presented in Table 3. The best-fitting solution is shown in Figure 8, which demonstrates that this model accurately captures the drying kinetics and is smoother than the shrinking-core model. The obtained criteria were $R^2 = 0.9678$ and RMSE $= 4.009 \times 10^{-6}$. In this model, $D_{eff}$ is time-dependent (Equation (18)). Appendix G presents a figure showing the evolution of $D_{eff}$ as a function of time, and the initial and final values for each experimental condition. These values correspond to the boundary positions and provide an accurate estimate of the order of magnitude because the $D_{eff}$ gradient in the product is small. At 40 °C, the initial $D_{eff}$ is equal to $8.957 \times 10^{-2}$ and reaches a final value of $3.425 \times 10^{-5}$ $m^2/s$. Similar orders of magnitude are obtained for the other experimental conditions. This shows that, at the beginning of the drying process, the capillary term (Equation (18)) is very high and gradually decreases until $D_{eff}$ falls within the expected range for diffusion-driven transport [36].

To further test the cooking model, the same adjustment steps were applied to all the experimental data sets, with one temperature condition excluded each time (leave-one-out fitting). The fittings remained accurate ($R^2 > 0.9488$). The results are shown in Appendix H.

### 3.3.4. Model Comparison

To compare the performance of the six models, the best-fitting parameters were used to simulate drying under all experimental conditions. The resulting model predictions were then evaluated against the experimental moisture content data $X$, temperature by temperature. Therefore, the $R^2$ and RMSE presented in Table 4 were calculated between the experimental and the predicted moisture content at each temperature.

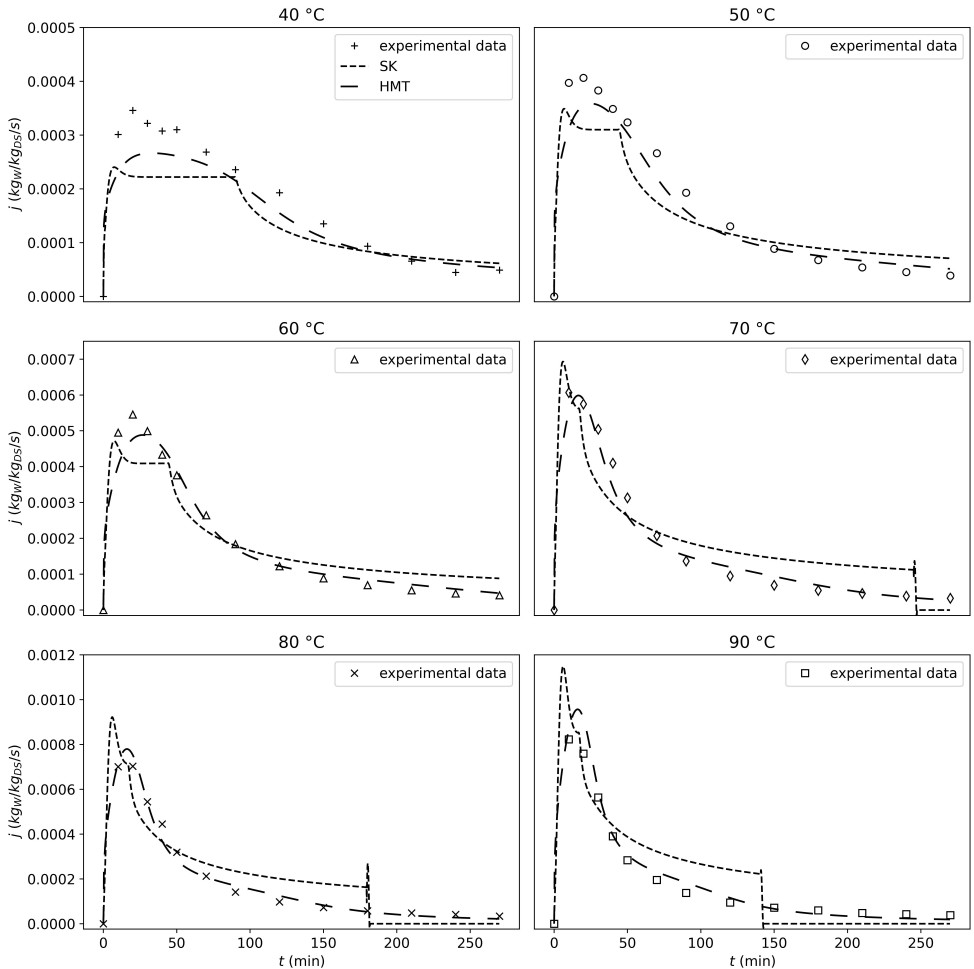

**Figure 8.** Fitting results for the shrinking-core (SK) and the cooking (HMT) models on the drying rate $j$ obtained from experimental data for drying temperatures 40 °C to 90 °C. Each experimental point is the mean of the triplicate.

The best-fitting model is the logarithmic model, followed by the Henderson–Pabis and the Page models. The cooking model has similar $R^2$ and RMSE values as the Page model, especially at intermediate temperatures (60 and 70 °C), for which the $R^2$ and RMSE values are the best. This is logical, as the parameters for the cooking model were fitted to all experimental curves simultaneously, while the parameters for the empirical models were fitted individually to each experimental condition.

The shrinking-core model has similar $R^2$ and RMSE values to the Newton model. However, unlike the Newton model, the shrinking-core model captures the allure of the experimental data (see Appendix F and Figure 8). The best-fitting results for the shrinking-core model are obtained at the three lower temperatures (40–60 °C). This can be explained by the allure of the fitted curves (see Figure 8): once a certain temperature is reached, the model compensates for the deviation from the experimental data by prematurely halting evaporation (the drying rate $j$ drops to 0).

**Table 3.** Parameters used in the cooking model for tofu drying.

| Parameter | Value or Correlation | Unit | Method or Source |
|:---:|:---:|:---:|:---:|
| $A$ | $-2.494$ | – | fitted |
| $B$ | $3.945$ | $kg_{DS}/kg_W$ | fitted |
| $Z$ | $8.261 \times 10^{-8}$ | $m^2/s$ | fitted |
| $\lambda_{DS}$ | $1.513 \times 10^{-1}$ | $W/m/K$ | fitted |
| $K_{BET}$ | $26.76$ | – | fitted |
| $X_{BET}$ | $0.632$ | $kg_W/kg_{DS}$ | fitted |
| $C_{p_{DS}}$ | $349.5 \times \left(1 - \frac{X_0}{1+X_0}\right)$ | $J/kg_{DS}/K$ | [12] |
| $D$ | $1.87 \times 10^{-10} \times \frac{T_{in}^{2.072}}{P}$ | $m^2/s$ | [35] |
| $h$ | $Nu = \frac{hL}{k} = 0.094 Re^{0.675} Pr^{1/3}$ | $W/m^2/K$ | [35] |
| $k_{HMT}$ | $Sh = \frac{kL}{D} = 0.094 Re^{0.675} Sc^{1/3}$ | $m/s$ | [35] |
| $C_A$ | $1.440 \times 10^{-3}$ | $kg_W/m_A^3$ | estimated |

**Table 4.** $R^2$ and RMSE values recalculated for the moisture content $X$ at every experimental drying temperature using the parameters obtained from the fitting of six models (Newton, Page, Henderson–Pabis (H-P), logarithmic (log), shrinking core (SK), and cooking (HMT)) to the drying rate obtained from experimental data.

| **$R^2$** | | | | | | |
|:---:|:---:|:---:|:---:|:---:|:---:|:---:|
| T (°C) | Newton | Page | H-P | Log | SK | HMT |
| 40 | 0.9792 | 0.9923 | 0.9933 | 0.9987 | 0.8552 | 0.9491 |
| 50 | 0.9314 | 0.9818 | 0.9905 | 0.9993 | 0.9312 | 0.9762 |
| 60 | 0.8911 | 0.9870 | 0.9884 | 0.9984 | 0.9624 | 0.9844 |
| 70 | 0.8249 | 0.9868 | 0.9878 | 0.9964 | 0.8257 | 0.9814 |
| 80 | 0.8108 | 0.9874 | 0.9883 | 0.9953 | 0.7290 | 0.9654 |
| 60 | 0.7426 | 0.9881 | 0.9887 | 0.9920 | 0.4408 | 0.8328 |

| **RMSE** | | | | | | |
|:---:|:---:|:---:|:---:|:---:|:---:|:---:|
| T (°C) | Newton | Page | H-P | Log | SK | HMT |
| 40 | $3.443 \times 10^{-2}$ | $2.100 \times 10^{-2}$ | $1.949 \times 10^{-2}$ | $8.725 \times 10^{-3}$ | $9.091 \times 10^{-2}$ | $5.390 \times 10^{-2}$ |
| 50 | $5.631 \times 10^{-2}$ | $2.237 \times 10^{-2}$ | $2.096 \times 10^{-2}$ | $5.665 \times 10^{-3}$ | $5.640 \times 10^{-2}$ | $3.318 \times 10^{-2}$ |
| 60 | $7.741 \times 10^{-2}$ | $2.672 \times 10^{-2}$ | $2.524 \times 10^{-2}$ | $9.479 \times 10^{-3}$ | $4.546 \times 10^{-2}$ | $2.927 \times 10^{-2}$ |
| 70 | $8.606 \times 10^{-2}$ | $2.365 \times 10^{-2}$ | $2.271 \times 10^{-2}$ | $1.233 \times 10^{-2}$ | $8.586 \times 10^{-2}$ | $2.808 \times 10^{-2}$ |
| 80 | $9.624 \times 10^{-2}$ | $2.483 \times 10^{-2}$ | $2.393 \times 10^{-2}$ | $1.517 \times 10^{-2}$ | $1.152 \times 10^{-1}$ | $4.117 \times 10^{-2}$ |
| 90 | $1.112 \times 10^{-1}$ | $2.394 \times 10^{-2}$ | $2.325 \times 10^{-2}$ | $1.957 \times 10^{-2}$ | $1.638 \times 10^{-1}$ | $8.959 \times 10^{-2}$ |

## 4. Discussion

The modeling results show that the Newton model can only capture one linear drying mechanism, which is insufficient for describing the drying kinetics of tofu. This model lacks physical validity and should not be used to describe tofu-drying kinetics.

The Page, Henderson–Pabis, and Logarithmic models show good results and fit the experimental data well.

As shown in Figure 8, the shrinking-core model presents good results. Although the mean criteria values ($R^2$ and RMSE) are no better than those of the Newton model, the drying mechanisms are captured much more accurately. This makes sense, as the model is capable of predicting two distinct drying periods (constant-rate and falling-rate). However, while both distinct periods are captured, the fitting accuracy is not satisfactory for all experimental conditions. Specifically, from 70 °C to 90 °C, the model poorly describes the falling-rate period, compensating by prematurely halting evaporation (i.e., the drying rate *j*

suddenly drops to zero, which has no physical validity). Therefore, this model could be used for control and optimization if the parameters were adapted to each experimental condition. However, it has no predictive or explanatory power, as fitting the parameters to all experimental conditions simultaneously does not produce satisfactory results for each condition, despite an overall satisfactory $R^2$.

The obtained value for $k_{SK}$ is of the same order of magnitude as the value obtained for $k_{HMT}$ using literature correlations (see Tables 2 and 3). In this model, porosity and tortuosity are considered constant, meaning that shrinkage is not taken into account. Porosity was calculated for the initial volume of 2 cm$^3$ and was found to be $0.86 \pm 0.05$, which is quite high but still within the range observed for other porous food products in the literature [37]. High porosity was also observed through visual examination of the sample, which means that tofu is a product with low structure. This explains why the product collapses significantly. From the calculated porosity and the fitted $\xi$, tortuosity could be deduced, which was equal to $4.95 \pm 0.31$. This value is quite high, but it is within the range of tortuosity determined for bread [38]. Finally, the effective diffusivity $D_{eff}$ calculated from the fitting results for the shrinking-core model ranged from $4.809 \times 10^{-6}$ to $6.537 \times 10^{-6}$. This is near the expected range for food [36]. Overall, the fitted parameters obtained for the shrinking core model are consistent with literature data.

The cooking model results in the best fit to the experimental data $j$. This is confirmed by the good results of the $R^2$ and RMSE values and by Figure 8, where it can be seen that the drying kinetics are captured very well. Compared to the shrinking-core model, this model's particularity is that it can predict a complex, multi-phasic drying process without breaking the model down into two periods.

Although the model is physics-based, some expressions remain empirical. The fitted parameters $A$, $B$, and $Z$ have no physical meaning. However, these parameters allow us to calculate the effective diffusivity $D_{eff}$. The calculated values of $D_{eff}$ fall outside the expected range for food products at the beginning of the drying process, but progressively approach the expected range as the process continues (see Appendix G) [36]. This is due to the capillary term that is taken into account in the expression of $D_{eff}$ (Equation (18)). Initially, water transport is heavily influenced by capillary transport, which progressively diminishes throughout the process. Next, $K_{BET}$ and $X_{BET}$ have physical interpretations. $K_{BET}$ reflects the strength of the interaction between the adsorbate and the surface. The fitted value for this constant is within the expected range for food, with a value of 26.76 [31]. $X_{BET}$ is the monolayer capacity. The fitted value ($X_{BET} = 0.632$) is about an order of magnitude higher than values reported in the literature for other food products [31]. Therefore, the suitability of the BET model to estimate the water content in the cooking model could be discussed, and other models could be used to replace this relationship. For example, the GAB model [31] could be used instead. Finally, the value obtained for the thermal conductivity of the dry product ($\lambda_{DS} = 1.513 \times 10^{-1}$ W/m/K) allows for the calculation of the thermal conductivity of the product before drying ($\lambda$ at $X = X_0$); the obtained value is $\lambda = 4.987 \times 10^{-1} \pm 5.109 \times 10^{-3}$, which is consistent with values reported in the literature for other food products [39].

These fitting results demonstrate that the empirical, the shrinking-core, and the cooking models each have their own strengths and weaknesses.

Except for the Newton model, the empirical models fit the experimental data well. However, they lack physical meaning, and they have no predictive or explanatory power. Unlike the physical models, these models cannot fit the drying rate $j$.

Although the shrinking-core model shows weaker fitting results, it has only two fitted parameters. This is a significant advantage compared to the cooking model, which has six fitted parameters.

It is important to note that both physical models underwent a stringent fitting procedure. Only one set of parameters was adjusted to fit all experimental conditions. Moreover, the experimental data used for the fitting procedure consisted of the drying rate values $j$, to ensure that all the drying mechanisms were well captured. Even though fitting on $X$ would have yielded better results, some physical phenomena, such as the distinct drying periods, are less pronounced in this data. This is why it was chosen to fit on $j$. Even with this strict fitting procedure, the physical models can compete with the results of the empirical models (Table 4).

Shrinkage is not taken into account in either of the models. Regarding experimental results, it would be a considerable improvement to take it into account for simulations.

## 5. Conclusions

In this study, the convective drying behavior of tofu samples was investigated across a range of drying temperatures (40–90 °C), using both an oven and a tunnel dryer. Experimental analysis of the tofu-drying process showed that it can be described as a sequence of different mechanisms (drying phases I and II separated by $X_{crit} = 2.56$ kg$_{\text{W}}$/kg$_{\text{DS}}$). These mechanisms are consistent across all studied temperatures, with higher temperatures resulting in faster drying.

Drying further, a second threshold could be defined at $X = 1.39$ kg$_{\text{W}}$/kg$_{\text{DS}}$, based on product shrinkage. Until that point, the shrinkage mechanisms were not temperature-dependent. However, once this threshold was reached, shrinkage became temperature-dependent and was more significant for lower drying temperatures (Figure 5). Similar observations were made for non-enzymatic browning at a very similar threshold (Figure 6).

It can be concluded that a faster drying process will reduce physical alterations of the product if the desired final water content is below $X = 1.39$ kg$_{\text{W}}$/kg$_{\text{DS}}$. Therefore, higher drying temperatures should be prioritized to avoid these alterations.

Six models were assessed against the experimental data. The Newton model failed to adequately represent the drying behavior of tofu. The Logarithmic, Henderson–Pabis, and Page models showed good fitting results but lack physical validity. In contrast, both the shrinking core and the cooking models showed good capacity to predict the drying rate, while capturing the physical mechanisms occurring during the drying process. While the shrinking-core model requires separate fits for each experimental condition to properly capture the drying processes, it remains useful due to its simplicity, as it relies on only two fitting parameters. The newly developed cooking model delivered more accurate predictions ($R^2 = 0.9678$). It successfully captured the various drying periods without segmenting the process into distinct rate expressions. This offers a more comprehensive and adaptable approach despite its reliance on six adjustable parameters.

Future research should evaluate the accuracy of the cooking model for larger-scale drying processes. Additionally, since the model was accurate for oven-drying processes, it should be tested at higher temperatures to replicate cooking conditions. Taking physical variations, especially shrinkage, into account would also be a considerable improvement. Further study of the nutritional and structural changes in tofu during drying is also required.

**Author Contributions:** Conceptualization, C.H. and F.D.; methodology, C.H., C.V.E. and F.D.; software, C.H. and C.V.E.; formal analysis, C.H., C.V.E. and F.D.; investigation, C.H. and M.A.H.; resources, C.H. and M.A.H.; data curation, C.H.; writing—original draft preparation, C.H. and F.D.; writing—review and editing, S.N., S.A. and C.V.E. ; visualization, C.H.; supervision, S.A. and F.D.; project administration, F.D.; funding acquisition, F.D. All authors have read and agreed to the published version of the manuscript.

**Funding:** The research leading to these results has been funded by the Public Service of Wallonia (Economy, Employment and Research), under the FoodWal agreement n° 2210182 from the Win4Excellence project of the Wallonia Recovery Plan.

**Data Availability Statement:** Experimental datasets will soon be uploaded and be available under https://doi.org/10.5281/zenodo.16949225.

**Acknowledgments:** We kindly thank Alice Wolper for her careful review of the present document and for her insightful remarks.

**Conflicts of Interest:** The authors declare no conflicts of interest. The funders had no role in the design of the study; in the collection, analyses, or interpretation of data; in the writing of the manuscript; or in the decision to publish the results.

## Nomenclature and Abbreviations

The following abbreviations are used in this manuscript:

Nomenclature

| | |
|---|---|
| $a$ | solid specific surface ($\text{m}^2_{\text{external surface}}/\text{kg}_{\text{DS}}$) |
| $a$ | empirical parameter |
| $a^*$ | redness |
| $A$ | empirical parameter |
| $A_W$ | water activity |
| $B$ | empirical parameter ($\text{kg}_{\text{DS}}/\text{kg}_{\text{W}}$) |
| $b^*$ | yellowness |
| $c$ | empirical parameter |
| $C$ | vapor mass concentration ($\text{kg}_{\text{W}}/\text{m}^3_{\text{A}}$) |
| $C_p$ | specific heat (J/kg/K) |
| $D$ | diffusion coefficient ($\text{m}^2/\text{s}$) |
| $h$ | heat transfer coefficient ($\text{W}/\text{m}^2/\text{K}$) |
| $j$ | drying rate ($\text{kg}_{\text{W}}/\text{kg}_{\text{DS}}/\text{s}$) |
| $k$ | mass transfer coefficient (m/s) |
| $k$ | kinetic constant ($\text{min}^{-1}$) |
| $K_{BET}$ | BET constant |
| $L$ | length of the sample (m) |
| $L^*$ | Lightness |
| $L_{vap}$ | latent heat of vaporization ($\text{J}/\text{kg}_{\text{W}}$) |
| $m$ | mass (kg) |
| $M$ | molar mass (kg/mol) |
| $n$ | empirical parameter |
| $N$ | number of experimental values |
| $P$ | pressure (Pa) |
| $Q$ | air mass flow rate ($\text{kg}_{\text{A}}/\text{s}$) |
| $R_g$ | ideal gas constant (J/mol/K) |
| $S$ | total shrinkage |
| $t$ | time (s) |
| $T$ | temperature (K) |
| $V$ | volume ($\text{m}^3$) |
| $X$ | water content in dry basis ($\text{kg}_{\text{W}}/\text{kg}_{\text{DS}}$) |
| $X_{BET}$ | monolayer moisture content ($\text{kg}_{\text{W}}/\text{kg}_{\text{DS}}$) |
| $x$ | spatial coordinate (m) |
| $Y$ | air water content in dry basis ($\text{kg}_{\text{W}}/\text{kg}_{\text{A}}$) |
| $Z$ | empirical parameter ($\text{m}^2/\text{s}$) |

Greek symbols

| | |
|---|---|
| $\alpha$ | volume fraction replaced by gaz |
| $\epsilon$ | porosity |
| $\lambda$ | thermal conductivity (W/m/K) |
| $\rho$ | volumetric mass (kg/m$^3$) |
| $\tau$ | tortuosity |
| $\Omega$ | surface area (m$^2$) |
| $\xi$ | porosity to tortuosity ratio |

Subscripts

| | |
|---|---|
| A | dry air |
| cap | capillary |
| DS | dry solid |
| eff | effective |
| exp | experimental |
| crit | critical |
| HMT | relative to the cooking model |
| $i$ | $i$th measure |
| in | inlet of drying oven |
| int | at interface |
| intern | internal |
| max | maximal value |
| pred | value predicted by the model |
| res | residual |
| sat | at saturation |
| SK | relative to the shrinking-core model |
| W | water |
| 0 | initial value |
| * | normalized value |
| I | constant drying rate period |
| II | falling drying rate period |

## Appendix A. Parameter Tables for the Shrinking Core and for the Cooking Model

Tables A1 and A2 show the parameter values or correlations used for the shrinking-core model and for the cooking model, respectively.

**Table A1.** Parameters used in the shrinking-core model for tofu drying.

| Parameter | Value or Correlation | Unit | Method or Source |
|---|---|---|---|
| $k_{SK}$ | $4.150 \times 10^{-3}$ | m/s | fitted |
| $\xi$ | $1.734 \times 10^{-1}$ | – | fitted |
| $a$ | $\frac{2.4 \times 10^{-3}}{m_{DS}}$ | m$^2_{\text{external surface}}$/kg$_{DS}$ | known |
| $C_{p_A}$ | 1005 | J/kg$_A$/K | known |
| $C_{p_W}$ | 4180 | J/kg$_W$/K | known |
| $L$ | $1 \times 10^{-2}$ | m | known |
| $L_{vap}$ | $2.26 \times 10^{-6}$ | J/kg$_W$ | known |
| $M_A$ | $2.9 \times 10^{-2}$ | kg$_A$/mol$_A$ | known |
| $M_W$ | $1.8 \times 10^{-2}$ | kg$_W$/mol$_W$ | known |
| $P$ | 101325 | Pa | known |
| $R_g$ | 8.314 | J/kg/K | known |
| $T_{in}$ | 40, 50, 60, 70, 80, 90 | °C | known |
| $T_{ref}$ | 100 | °C | known |
| $V_{oven}$ | $5.3 \times 10^{-2}$ | m$^3$ | known |

**Table A1.** *Cont.*

| Parameter | Value or Correlation | Unit | Method or Source |
|---|---|---|---|
| $\rho_A$ | 1.2 | $kg_A/m_A^3$ | known |
| $X_{crit}$ | 2.56 | $kg_W/kg_{DS}$ | experimental |
| $C_{sat}$ | $\frac{PM_W}{R_gT}\exp\left(-\frac{L_{vap}M_W}{R_g}\left(\frac{1}{T}-\frac{1}{T_{ref}}\right)\right)$ | $kg_W/m_A^3$ | calculated |
| $D$ | $1.87\times10^{-10}\times\frac{T_{in}^{2.072}}{P}$ | $m^2/s$ | [35] |
| $C_{p_{DS}}$ | $349.5\times\left(1-\frac{X_0}{1+X_0}\right)$ | $J/kg_{DS}/K$ | [12] |
| $Q$ | $2\times10^{-4}$ | $kg_A/s$ | estimated |
| $X_{res}$ | 0.00 | $kg_W/kg_{DS}$ | estimated |
| $Y_{in}$ | $1\times10^{-3}$ | $kg_W/kg_A$ | estimated |

**Table A2.** Parameters used in the cooking model for tofu drying.

| Parameter | Value or Correlation | Unit | Method or Source |
|---|---|---|---|
| $A$ | $-2.494$ | – | fitted |
| $B$ | $3.945$ | $kg_{DS}/kg_W$ | fitted |
| $Z$ | $8.261\times10^{-8}$ | $m^2/s$ | fitted |
| $\lambda_{DS}$ | $1.513\times10^{-1}$ | $W/m/K$ | fitted |
| $K_{BET}$ | $26.76$ | – | fitted |
| $X_{BET}$ | $6.320\times10^{-1}$ | $kg_W/kg_{DS}$ | fitted |
| $C_{p_W}$ | $4180$ | $J/kg_W/K$ | known |
| $L$ | $2\times10^{-2}$ | m | known |
| $T_{in}$ | 40, 50, 60, 70, 80, 90 | °C | known |
| $V$ | $8\times10^{-6}$ | $m^3$ | known |
| $\Omega$ | $4\times10^{-4}$ | $m^2$ | known |
| $C_{p_{DS}}$ | $349.5\times\left(1-\frac{X_0}{1+X_0}\right)$ | $J/kg_{DS}/K$ | [12] |
| $D$ | $1.87\times10^{-10}\times\frac{T_{in}^{2.072}}{P}$ | $m^2/s$ | [35] |
| $h$ | $Nu=\frac{hL}{k}=0.094Re^{0.675}Pr^{1/3}$ | $W/m^2/K$ | [35] |
| $k_{HMT}$ | $Sh=\frac{kL}{D}=0.094Re^{0.675}Sc^{1/3}$ | m/s | [35] |
| $C_A$ | $1.440\times10^{-3}$ | $kg_W/m_A^3$ | estimated |

## Appendix B. Moisture Content Evolution as a Function of Oven Drying Temperature

Figure A1 shows the experimental data obtained from the convective drying oven experiments. Each experimental condition was tested in triplicate, and the figure shows that the results are reproducible.

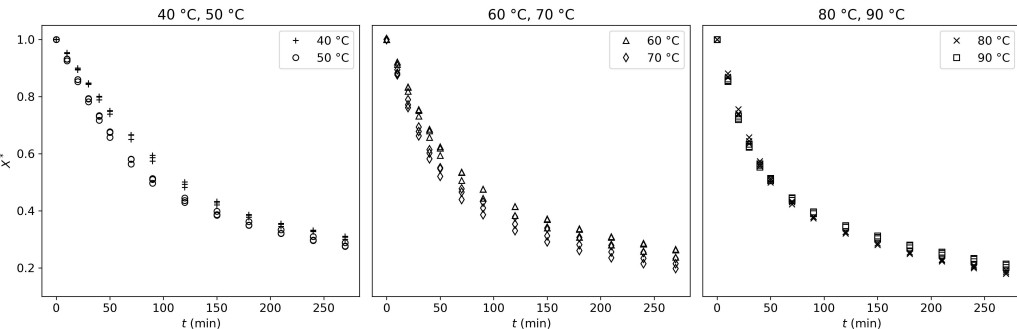

**Figure A1.** Evolution of the relative moisture content *X\** for different convective oven-drying temperatures (40 °C to 90 °C).

## Appendix C. Tunnel and Oven Drying Comparison

The tunnel-drying experiments were compared to the oven-drying experiments conducted at the same temperatures (Section 3.1). This is shown in Figure A2, where the evolution of the relative drying rate $j^*$ for the two drying methods is compared.

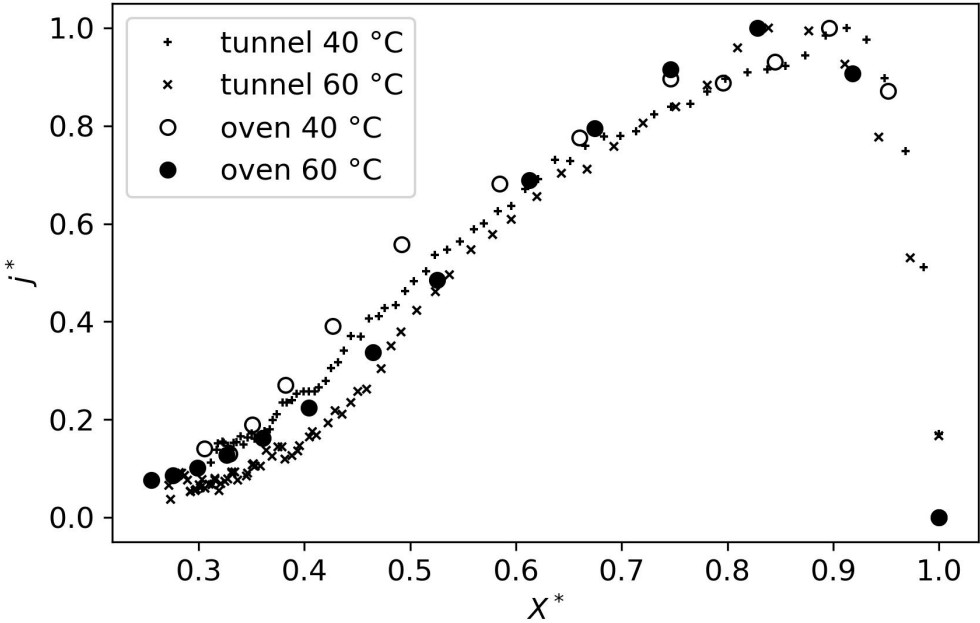

**Figure A2.** Evolution of the relative drying rate $j^*$ of tofu samples as a function of the relative moisture content $X^*$ for convective drying experiments in a drying oven and in a drying tunnel at two drying temperatures.

## Appendix D. Fitting of the Shrinkage Model to Experimental Results

The shrinkage of tofu during drying was compared to the model described in Equation (7). The parameter $\alpha$ was adjusted to fit the normalized volume $V^*$ calculated from each experimental dataset (Section 3.2). Figure A3 shows the fitting results for the linear shrinking period for one experiment per drying temperature.

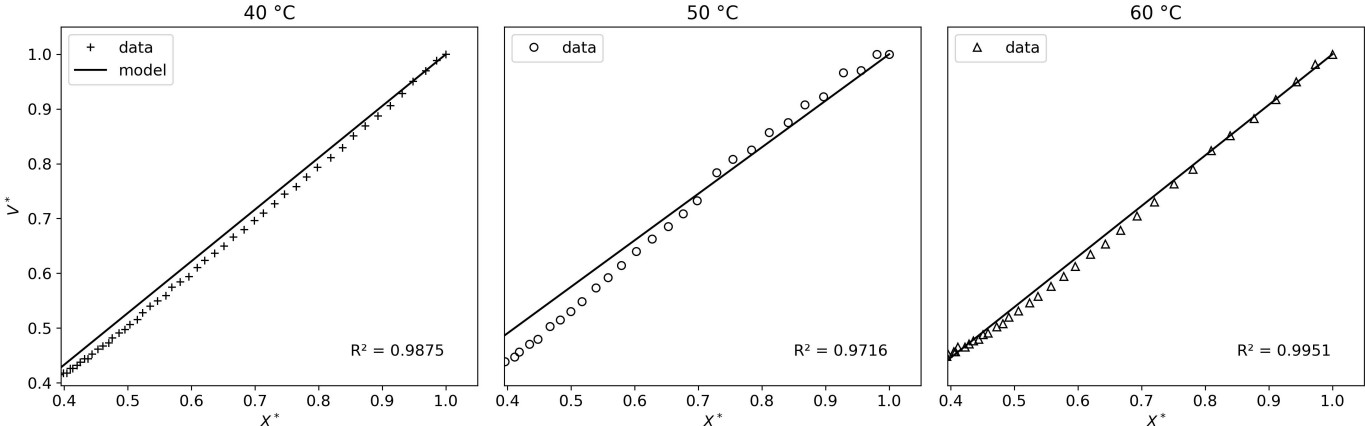

**Figure A3.** Evolution of the normalized volume $V^*$ of the tofu samples as a function of relative water content $X^*$ during convective drying experiments at three different temperatures and fitting of a linear model to the experimental data.

## Appendix E. Color Variation Results

This appendix presents the results obtained for each color component for the color variation analysis during drying (Section 3.2). All three components $L^*$, $a^*$, and $b^*$ are presented as a function of time and as a function of relative moisture content in Figure A4.

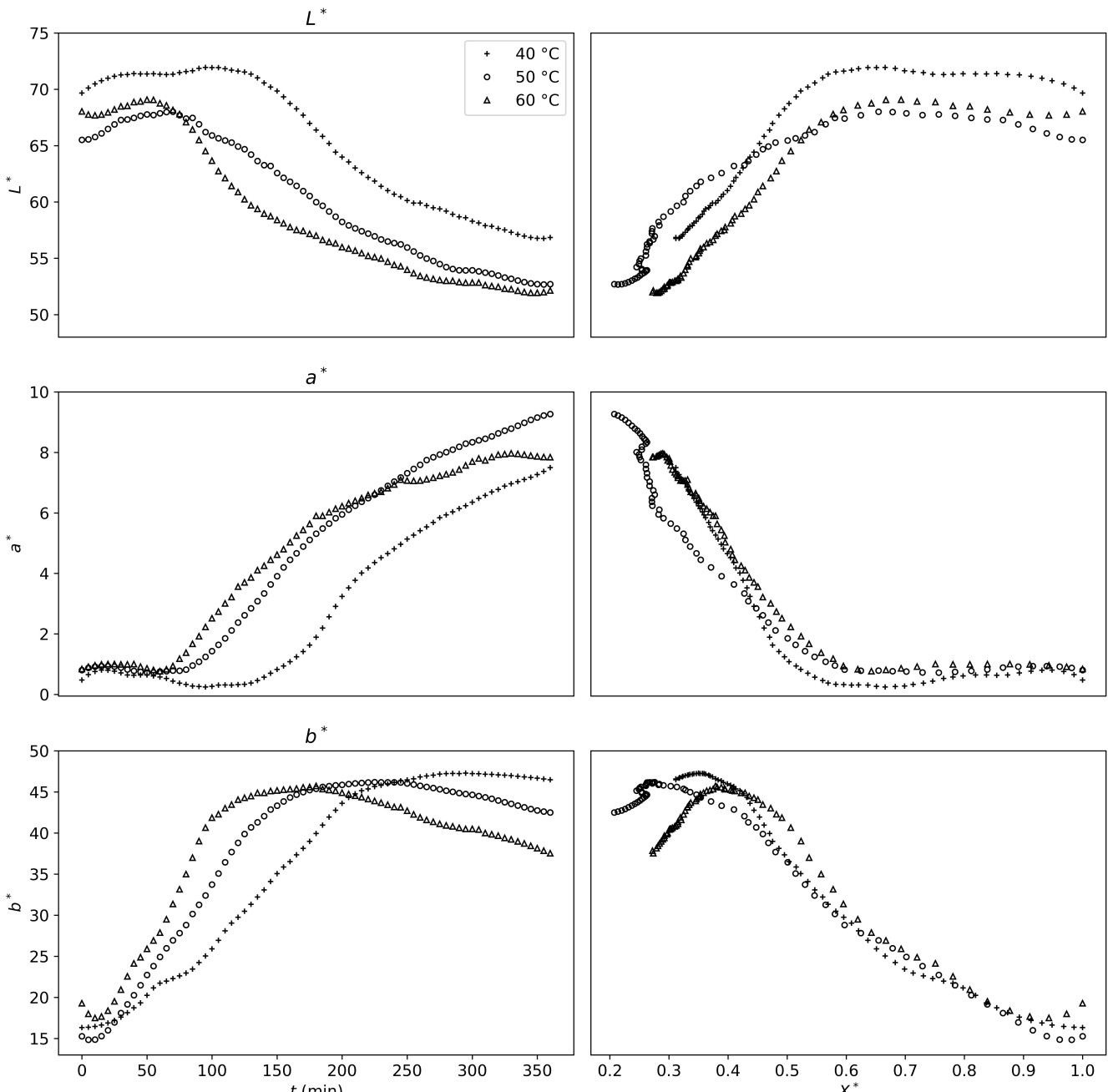

**Figure A4.** Evolution of lightness index $L^*$ (**first row**), redness index $a^*$ (**second row**), and yellowness index $b^*$ (**third row**) from the CIELAB color space as a function of time (**left**) and as a function of relative moisture content $X^*$ (**right**) for tofu samples during convective drying at different temperatures.

## Appendix F. Empirical Models Results

Table A3 shows the identified values of the parameters for the four empirical models (Section 3.3), and the corresponding values of $R^2$ and RMSE (between the model and the experimental $X^*$ values). The fitting results are shown in Figure A5.

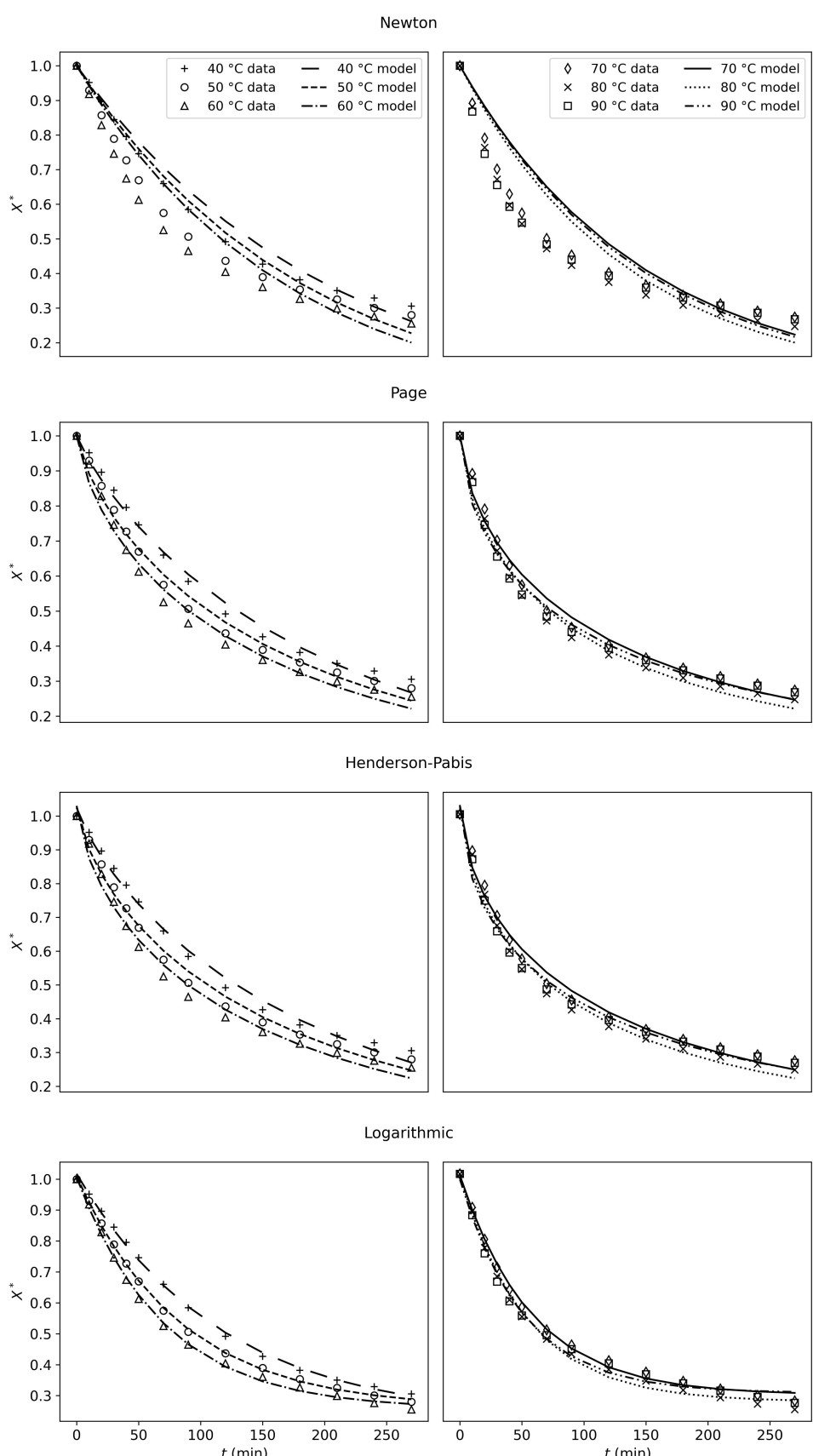

**Figure A5.** Fitting of the four empirical models (Newton, Page, Henderson–Pabis, and logarithmic) to experimental drying data at different drying temperatures.

**Table A3.** Adjusted parameter-values for the four empirical models (Newton, Page, Henderson–Pabis, and logarithmic), and $R^2$ and RMSE values obtained from the fitting to each dataset.

| | Newton | | | | |
|---|---|---|---|---|---|
| T (°C) | $k$ (min$^{-1}$) | | | $R^2$ | RMSE |
| 40 | $4.963 \times 10^{-3}$ | | | 0.9676 | $1.966 \times 10^{-2}$ |
| 50 | $5.488 \times 10^{-3}$ | | | 0.9209 | $3.199 \times 10^{-2}$ |
| 60 | $5.955 \times 10^{-3}$ | | | 0.8861 | $4.040 \times 10^{-2}$ |
| 70 | $6.785 \times 10^{-3}$ | | | 0.8366 | $5.304 \times 10^{-2}$ |
| 80 | $7.415 \times 10^{-3}$ | | | 0.8321 | $5.840 \times 10^{-2}$ |
| 60 | $6.957 \times 10^{-3}$ | | | 0.7812 | $6.057 \times 10^{-2}$ |

| | Page | | | | |
|---|---|---|---|---|---|
| T (°C) | $k$ (min$^{-1}$) | $n$ | | $R^2$ | RMSE |
| 40 | $9.653 \times 10^{-3}$ | $8.785 \times 10^{-1}$ | | 0.9923 | $5.604 \times 10^{-3}$ |
| 50 | $1.982 \times 10^{-2}$ | $7.616 \times 10^{-1}$ | | 0.9892 | $6.628 \times 10^{-3}$ |
| 60 | $2.810 \times 10^{-2}$ | $7.114 \times 10^{-1}$ | | 0.9870 | $7.346 \times 10^{-3}$ |
| 70 | $4.309 \times 10^{-2}$ | $6.553 \times 10^{-1}$ | | 0.9868 | $7.523 \times 10^{-3}$ |
| 80 | $4.821 \times 10^{-2}$ | $6.516 \times 10^{-1}$ | | 0.9874 | $7.544 \times 10^{-3}$ |
| 60 | $5.998 \times 10^{-2}$ | $5.956 \times 10^{-1}$ | | 0.9881 | $6.927 \times 10^{-3}$ |

| | Henderson–Pabis | | | | |
|---|---|---|---|---|---|
| T (°C) | $k$ (min$^{-1}$) | $n$ | $a$ | $R^2$ | RMSE |
| 40 | $1.239 \times 10^{-2}$ | $8.355 \times 10^{-1}$ | 1.024 | 0.9933 | $5.202 \times 10^{-3}$ |
| 50 | $2.468 \times 10^{-2}$ | $7.246 \times 10^{-1}$ | 1.028 | 0.9905 | $6.212 \times 10^{-3}$ |
| 60 | $3.424 \times 10^{-2}$ | $6.783 \times 10^{-1}$ | 1.030 | 0.9884 | $6.939 \times 10^{-3}$ |
| 70 | $4.973 \times 10^{-2}$ | $6.316 \times 10^{-1}$ | 1.027 | 0.9878 | $7.227 \times 10^{-3}$ |
| 80 | $5.482 \times 10^{-2}$ | $6.302 \times 10^{-1}$ | 1.026 | 0.9883 | $7.270 \times 10^{-3}$ |
| 60 | $6.644 \times 10^{-2}$ | $5.790 \times 10^{-1}$ | 1.022 | 0.9888 | $6.728 \times 10^{-3}$ |

| | Logarithmic | | | | |
|---|---|---|---|---|---|
| T (°C) | $k$ (min$^{-1}$) | $a$ | $c$ | $R^2$ | RMSE |
| 40 | $8.608 \times 10^{-3}$ | $7.973 \times 10^{-1}$ | $2.199 \times 10^{-1}$ | 0.9987 | $2.330 \times 10^{-3}$ |
| 50 | $1.192 \times 10^{-2}$ | $7.508 \times 10^{-1}$ | $2.582 \times 10^{-1}$ | 0.9993 | $1.679 \times 10^{-3}$ |
| 60 | $1.415 \times 10^{-2}$ | $7.492 \times 10^{-1}$ | $2.564 \times 10^{-1}$ | 0.9984 | $2.613 \times 10^{-3}$ |
| 70 | $1.725 \times 10^{-2}$ | $7.589 \times 10^{-1}$ | $2.354 \times 10^{-1}$ | 0.9964 | $3.953 \times 10^{-3}$ |
| 80 | $1.848 \times 10^{-2}$ | $7.798 \times 10^{-1}$ | $2.116 \times 10^{-1}$ | 0.9953 | $4.599 \times 10^{-3}$ |
| 60 | $1.960 \times 10^{-2}$ | $7.379 \times 10^{-1}$ | $2.425 \times 10^{-1}$ | 0.9920 | $5.676 \times 10^{-3}$ |

# Appendix G. $D_{eff}$ Values and Evolution in the Cooking Model

In the cooking model, $D_{eff}$ is time-dependent (Equation (18)). Table A4 shows the initial and final values of $D_{eff}$ for each drying condition and Figure A6 shows the evolution of $D_{eff}$ with time as predicted by the model. The values presented correspond to the boundary positions of the product and give a good idea of the order of magnitude of $D_{eff}$ in the product, as the gradient in the product is small.

**Table A4.** $D_{eff}$ values at the beginning and at the end of the drying process for each drying condition, as predicted by the cooking model.

| Drying Temperature | $D_{eff_0}$ (m$^2$/s) | $D_{eff_{final}}$ (m$^2$/s) |
|---|---|---|
| 40 °C | $8.957 \times 10^{-2}$ | $3.425 \times 10^{-5}$ |
| 50 °C | $2.084 \times 10^{-2}$ | $3.118 \times 10^{-5}$ |
| 60 °C | $5.365 \times 10^{-2}$ | $3.220 \times 10^{-5}$ |
| 70 °C | $7.794 \times 10^{-3}$ | $3.378 \times 10^{-5}$ |
| 80 °C | $1.462 \times 10^{-2}$ | $3.574 \times 10^{-5}$ |
| 90 °C | $2.746 \times 10^{-2}$ | $3.782 \times 10^{-5}$ |

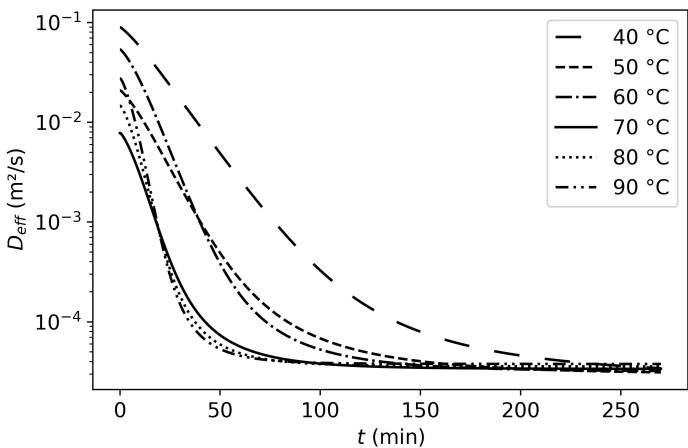

**Figure A6.** Evolution of $D_{eff}$ at different drying temperatures as predicted by the cooking model.

## Appendix H. Additional Fitting Method for the Cooking Model

To challenge the cooking model (Section 3.3) further, the fitting process was applied to the datasets, leaving one out: for each fitting, one of the experimental conditions is left out of the parameter adjustment process. Results are shown in Table A5.

**Table A5.** Fitting results for the cooking model when one of the experimental conditions is left out of the parameter adjustment.

| Left out Temperature | $R^2$ | RMSE |
|---|---|---|
| none | 0.9678 | $4.001 \times 10^{-6}$ |
| 40 °C | 0.9673 | $4.041 \times 10^{-6}$ |
| 50 °C | 0.9662 | $4.108 \times 10^{-6}$ |
| 60 °C | 0.9663 | $4.102 \times 10^{-6}$ |
| 70 °C | 0.9641 | $4.234 \times 10^{-6}$ |
| 80 °C | 0.9596 | $4.491 \times 10^{-6}$ |
| 90 °C | 0.9488 | $5.054 \times 10^{-6}$ |

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
