# Peer review of "Modeling and Experimental Analysis of Tofu-Drying Kinetics"

_applsci, doi:10.3390/app151910319_

Round 1
Reviewer 1 Report
Comments and Suggestions for Authors
1. The abstract is too long and does not highlight the key points. Most of it is qualitative description, lacking quantitative description and data support. It is recommended to rewrite it.
2. L31, Is this two paragraphs or one?
3. L36, There should be references instead of "as far as we know".
4. What is the relationship between soybeans, soybean residue, and tofu drying?
5. Why does tofu need to be dried? After reading the introduction, I did not feel the significance of this work. The introduction needs to be rewritten.
6. What is the reason for setting the drying temperature at L84? What is the basis?
7. Figure 1 is too simple. Are there any real pictures?
8. Has the model establishment considered the shrinkage and deformation of tofu? Will this affect the model?
9. How are the two drying stages divided? It needs to be written clearly.
10. There are few references in the last three years in the reference list. It is recommended to update it.
Reviewer 2 Report
Comments and Suggestions for Authors
applsci-3873850; Modeling and Experimental Analysis of Tofu Drying Kinetics
This manuscript contributes significantly to the study of tofu drying kinetics by integrating experimental methods—such as oven and tunnel drying—with mathematical modeling approaches, including the Newton model, a shrinking core model, and a novel HMT model. This research is original, as tofu drying has not been systematically examined before, providing valuable insights for academia and industry.
The study addresses an underexplored area in food drying and employs a solid experimental design that includes replicates and complementary drying systems. The shrinkage and color changes analysis effectively connects physical properties to drying kinetics. Furthermore, comparing different modeling approaches, especially with the cooking model, establishes a robust predictive framework. Identifying critical moisture thresholds for shrinkage and color change is of practical significance for industry applications.
The paper is well-organized, with clear objectives, a strong methodology, and comprehensive results. However, some sections require revision for improved clarity and conciseness.
Suggestions:
While the manuscript is understandable, it includes several long, complex sentences, hindering readability. I suggest using shorter and clearer sentences. Additionally, there are minor typographical and formatting inconsistencies, especially in equations and decimal formatting.
The choice of tofu type (firm, commercial product) is explained, but providing more information on its proximate composition (including protein, fat, and carbohydrate content) would enhance the study.
The section on the Newton model needs critical revision. Although the fitting results show relatively high R² values, the model fails to capture the drying mechanisms of tofu. The fitted kN values (0.0050–0.0074 min⁻¹) are one to two orders of magnitude lower than those typically reported for other foods, suggesting that the Newton model underestimates drying kinetics. The authors correctly identify this issue, but the discussion should be clearer: while the Newton model yields formally correct fittings, it lacks physical validity. The authors should also consider including other standard empirical drying models, such as the Page, Henderson–Pabis, or logarithmic models, often outperforming the Newton model in representing non-linear drying behavior. Including these models would strengthen the comparison and enhance the scientific robustness of the study.
Moreover, several sections of the results are overly descriptive and repeat numerical values already presented in tables and figures. For example, lines 238–249 repeat information visible in Figure 3; lines 269–279 duplicate shrinkage data from Figure 5; lines 299–309 reiterate color data from Figure 6 and Appendix E; and lines 331–337 restate Newton model fitting values already in Table A3. These should be shortened, focusing on interpretation and mechanistic explanation (e.g., porosity collapse for shrinkage and Maillard browning for color).
In the Conclusions section (lines 471–489), data already presented in the Results are repeated (critical moisture thresholds, color, and shrinkage). This section should be abbreviated to emphasize only the key take-home messages.
In summary, the manuscript is scientifically sound and relevant for publication in Applied Sciences. I suggest Major Revision
Round 2
Reviewer 1 Report
Comments and Suggestions for Authors
Accept in present form.